# Joining of Electrodes to Ultra-Thin Metallic Layers on Ceramic Substrates in Cryogenic Sensors [note 1]

**DOI:** 10.3390/s21144919

**Published:** 2021-07-20

**Authors:** Marcin Lebioda, Ryszard Pawlak, Jacek Rymaszewski

**Affiliations:** Institute of Electrical Engineering Systems, Lodz University of Technology, 90-924 Lodz, Poland; ryszard.pawlak@p.lodz.pl (R.P.); jacek.rymaszewski@p.lodz.pl (J.R.)

**Keywords:** soldering, thin layers, cryogenic sensors, indium, ceramics

## Abstract

Microjoining technologies are crucial for producing reliable electrical connections in modern microelectronic and optoelectronic devices, as well as for the assembly of electronic circuits, sensors, and batteries. However, the production of miniature sensors presents particular difficulties, due to their non-standard designs, unique functionality and applications in various environments. One of the main challenges relates to the fact that common methods such as reflow soldering or wave soldering cannot be applied to making joints to the materials used for the sensing layers (oxides, polymers, graphene, metallic layers) or to the thin metallic layers that act as contact pads. This problem applies especially to sensors designed to work at cryogenic temperatures. In this paper, we demonstrate a new method for the dynamic soldering of outer leads in the form of metallic strips made from thin metallic layers on ceramic substrates. These leads can be used as contact pads in sensors working in a wide temperature range. The joints produced using our method show excellent electrical, thermal, and mechanical properties in the temperature range of 15–300 K.

## 1. Introduction

The quality of electronic connections has a decisive impact on the reliability of electronic and mechatronic devices. The continuing development and miniaturization of electronic devices has created new challenges related to electrical connections. The factors causing these difficulties can be divided into three groups: technological factors, environmental factors, and functional factors. Technological factors include packaging density (the miniature dimensions of the elements to be joined, small contact pads, very thin wires), dimensional disproportions (e.g., connecting a thin wire to a massive element) and the physical properties of the materials (no metallurgical affinity, lack of solderability or weldability). Environmental factors include operating conditions such as humidity, extreme temperature, strain, vibration, or other harsh conditions. Functional factors relate to the increased requirements for reliability (for military technology, aviation, spacecraft, automobiles, and safety systems).

Microjoining technologies are crucial for fabricating modern microelectronic devices (e.g., integrated circuits (ICs)) and electronic circuits, but also for sensors, batteries, and optoelectronics. In manufacturing, methods specific to a given assembly level are used to produce electrical connections in electronic devices. At the first level, electrical connections are made between the semiconductor structure and the external terminals of the IC packaging. Flip-chip soldering [1] using lead-free solders or conductive adhesives [2] is commonly used for assembling integrated circuit structures [3] microsystems (MEMS) structures [4] and optoelectronic devices [5]. At the second level, wire bonding is used to join thin Au wire to a contact pad (Au or Al) on a semiconductor structure using thermocompression, or thin primarily Al wire to a contact pad (Au or Al) using ultracompression [3]. To reduce the costs of Au wire bonding, copper wire bonding was developed [6]. Producing connections on the second assembly level depends on the type of housing that is used for the electronic elements. Integrated circuits in a pin grid array (PGA) or dual in-line (DILP) housing are suitable for through-hole assembly by means of soldering. Most miniature individual elements, such as encapsulated resistors, capacitors, diodes, transistors, and ICs in a quad flat package (QFP), are suitable for surface mounting on PCB (printed boards) using soldering with lead-free solder. Microprocessors with many hundreds of pins arranged as a ball grid array (BGA) are suitable for surface soldering [7]. In addition to soldering, there are many other technologies involved in the process of microjoining, including solid state bonding, fusion microwelding, solid-state diffusion bonding, diffusion soldering and brazing, resistance microwelding and adhesive bonding [8]. Laser technologies are also applied to produce different types of electrical joints [9,10,11,12].

Soldering technology is used to make electrical and mechanical connections between electronic components that work across a wide temperature range. A joint made by soft soldering should be characterized by good and stable electrical conductivity, as well as tightness and the required mechanical strength, which should be resilient during storage and operation. In the assembly of consumer electronic devices such as computers, smartphones, TVs and device control systems, batch processes such as reflow soldering or wave soldering are commonly used. In all soldered joints growth kinetics of intermetallic phases is of great importance [13]. Recently, efforts have been focused on the development of methods for producing durable joints in energy storage devices [14,15]. In MEMS packaging, it is necessary to use progressively lower temperatures for joining (bonding) in the first, second and final level of the assembly process. The best solutions are then transient liquid phase bonding (TLP) or diffusion soldering [16]. The joints made using TLP can survive temperatures higher than their formation temperature, so this method is used in the assembly of thermoelectric power modules and power devices [17,18].

New problems related to the production of electrical connections have emerged with the development of flexible and stretchable electronics [19,20,21]. Producing durable and reliable joints between flexible materials and outer leads requires special technological processes, although common methods including soldering and adhesive joining with nanoparticles are also used [22]. Thin metallic layers are often used as conducting paths or contact pads in stretchable electronic devices. The use of standard soldering technology to make connections with such thin metallic layers is especially problematic. Producing electrical connections involving thin sensing layers made from non-metallic materials (transparent conducting oxides [23,24], polymers, graphene [25,26], metallic nanowires layers, layers of carbon nanotubes [27]) is also challenging.

Due to miniaturization, sensors usually do not have the same construction as standard semiconductor devices, and the outer leads are connected directly to the sensing layer or thin metallic pad. Problems with electrical connections occur especially in sensors working at cryogenic temperatures [23,24,25,26,28]. Such connections must operate over a wide temperature range and are therefore subject to extreme thermal stress. The geometry of the joined elements and the materials from which they are made result in unequal stresses in the joint area. It is extremely important to select materials and technologies for the connection that can ensure reliable operation under cryogenic conditions. Indium and indium alloys are commonly used for the production of joints designed to work at cryogenic temperatures, as indium remains highly susceptible to plastic deformation at low temperatures, and therefore eliminates high stresses at the boundary of joined elements composed of different materials. This approach has been used, for example, to join tapes made from 2G HTS (a second generation high temperature superconductor) [29].

The present work set out to solve two main issues: (1) how to produce a joint between an outer lead (metal strip) and thin metallic layer without causing damage to the metallic layer, and (2) how to fulfill the requirements for connections operating at cryogenic temperatures. The decisive property of a good joint is its electrical resistance, which results from the geometry and metallurgical properties of the joint, i.e., the type and morphology of the phases. Of particular importance is the occurrence of intermetallic phases at the interfaces between the thin layer of metallization and the material of the joint, as well as between the metal strip (outer lead) and the material of the joint. The presence of intermetallic phases contributes to the mechanical strength of the joint, but in general increases its electrical resistance. Another question, then, was how to produce a joint with low electrical resistance where the area of the joint is not the dominant heat source. The heat generated in the electrical joints heats up the sensor (self-heating effect) and is a continuous additional heat load of the cooling system. The experimental and simulation studies indicating the dominant source of heat generated as a result of electric current flow were performed. Connections with such properties and geometry cannot be made using current methods (e.g., thermocompression). Directly heating the solder (indium) on the surface of the metallic layer with a stream of hot air (200 °C) also gives unsatisfactory results (see Section 2.1).

The new method presented here enables a permanent soldered joint to be made between Ag and Au electrodes and a thin metallic layer (Ag, Au) on a ceramic substrate (Al_2_O_3_). We reported good preliminary results using our new method to join outer lead (Ag strip) to Au metallic layer in a conference paper [30]. This manuscript presents a much expanded study, including: strength tests of the joints; a study of the electrical and thermal properties of the joints at room temperature; development of a thermo-electrical model of the joint that enables the analysis of thermal phenomena and temperature effects on the electrical properties of materials; detailed metallographic studies of the joints, including analysis of the phase composition of the joint area; an assessment of the possibility of using our new method to make connections between other materials [30].

## 2. Materials and Methods

### 2.1. New Method of Soldering to Thin Metallic Layers

The presented joining method is applicable to realize various types of sensors operating at cryogenic temperatures. It was tested on the two metals most often used in the construction of sensor terminals and electrodes: gold and silver. Gold is a suitable material for electrodes in sensors [26]. The use of silver for outer leads facilitates integration with the measuring system and reduces the resistance of the leads. We conducted microscopic studies on various sample configurations to analyze the intermetallic compounds formed in the joint. Our main focus, however, was on the electrical and mechanical properties of the joint between a silver strip and thin gold metallization on a ceramic substrate.

An ultra-thin metallic layer with a thickness of 200 nm was deposited on an 800 µm thick Al_2_O_3_ ceramic substrate. The metallic layer was made using the PVD (Physical Vapor Deposition) method with a Classic 250 vacuum system (Pfeiffer Vacuum, Annecy, France). A 2 mm wide metal strip was used as an outer flexible lead. Indium was used as the solder. The samples configuration are presented in Table 1.

Briefly, our method consists of indirectly heating and melting solder on a ceramic substrate coated with a metallic layer, and then introducing a metal strip with a gel flux to facilitate the soldering of each surface. Immediately after the strip with flux has been introduced, the joint cooling process begins. The whole process requires only a very short time. The three-step process is presented in Figure 1.

In the first step of the soldering process, solder is placed on a ceramic substrate covered with a metallic layer, which is then heated (Figure 1a). The solder is used in solid form, preferably as a small ball. Solder balls were made from pieces of thin In wire placed on a polished ceramic substrate and melted in a vacuum. This is a separate process, unrelated to the soldering shown in Figure 1. The spherical shape reduces the contact area of the solder with the substrate and eliminates the possibility of uncontrolled wetting of the metallized surface by the solder. Uncontrolled wetting could lead to damage to the metal layer, due to surface tension in the liquid solder. The solder is heated and melted indirectly by a stream of hot air (~200 °C) directed at the underside of the ceramic substrate. Direct heating with a stream of hot air would move the melted solder over the surface of the substrate and damage the layer. Heating with hot air from the bottom also eliminates the possibility of damaging the metal layer with a hot tip. The absence of flux at this stage of the process is crucial, as it prevents the solder from spreading over the metal layer which remains continuous and undamaged. Importantly, the process of heating the substrate is volumetric, which means that it facilitates the spread and penetration of the solder in the next step of joint formation. Figure 2a and b show the results when solder and flux are introduced simultaneously. The layer is permanently damaged and the damage spreads when heated. Figure 2c shows the result of heating the solder (indium) without flux. It should be noted that the solder does not wet the surface and remains almost spherical. In the presented cases, the substrate was heated for 10 s from the moment the solder was melted.

In the second step (Figure 1b), a metal strip with a flux coating on the joined area of the strip is placed over the melted solder. This is a crucial step in our method. The flux ensures precise dosing and controlled spreading. The flux changes the surface tension of the solder and initiates instant wetting of both joined surfaces. Due to strong adhesion, the solder is trapped between the joined surfaces. The solder spreads very quickly and evenly, in the form of a thin layer between the joined elements. The thin layer of metal on the ceramic is neither destroyed nor delaminated, and intermetallic compounds are formed in the joint volume.

The third and final step in the soldering process (Figure 1c) is cooling. It should be noted that the delay between the introduction of the strip with the flux and the beginning of the cooling stage is very short, around 1–2 s. This is important because the time and thermal parameters are the main determinants of the structure of the final joint.

### 2.2. Instrumentation and Measurement Procedure

The structure of the joints was investigated by means of optical microscopy (Neophot 21, Carl-Zeiss Jena, Germany) and scanning electron microscopy (JEOL JSM–IT200, JEOL Ltd., Akishima, Tokyo, Japan), using metallographic sections. The phase compositions of the layers of the joints were investigated using a microanalysis system (JED-2300/2300F, JEOL Ltd., Akishima, Tokyo, Japan).

The electrical parameters of the joints were measured at room temperature and at cryogenic temperature. The studies at cryogenic temperatures were performed using a helium closed-cycle DE-210 cryostat (Advanced Research Systems, Inc., Macungie, PA, USA) with a Lake Shore 331 temperature controller (Lakeshore Cryotronics Inc., Westerville, OH, USA). The sample was placed in a vacuum chamber and cyclically cooled and heated over a temperature range of 300–15 K at a rate of about 4 K/min. The lower surface of the ceramic was attached to a copper heat exchanger to eliminate the temperature gradient in the substrate. The reference temperature sensor (silicone diode DT-670-SD Lakeshore Cryotronics Inc., Westerville, OH, USA) was mounted directly on the heat exchanger next to the tested sample. A massive copper heat exchanger was mounted directly to the “cold finger” of the cryocooler. A Keysight 34420A Micro-Ohm meter (Keysight Technologies, Santa Rosa, CA, USA) was used to measure the resistance of the tested samples, according to the four-probe method. The resistance of the joint and the resistance of the metallic layer were measured simultaneously. Figure 3 shows the arrangement of the electrodes and the measurement method used.

The resistance *R_L_* of the length *D* of the metal layer was measured. The resistance of the section consisting of the layer (*D*/2) and the joint (*D*/2) was measured at the same time (Figure 3). This enabled us to determine the temperature dependence of the joint resistance *R_J_* (*T*) and the layer resistance *R_L_* (*T*). We assumed that the Au layer was continuous over the measured section.

When measuring the electrical parameters of the samples at room temperature, we focused on the Joule heat flux density generated by the current in samples with the same geometry. The aim was to identify the dominant heat source. If the joint has low resistance, then the area of the joint is not the dominant heat source. Small thermocouples (T-type, wire diameter 140 µm) were attached to the surface of the sample at its center (T_c_) and at the joint (T_j_). The temperature was measured at different currents under conditions of natural air convection (Figure 4).

Current was applied to the sample from a controlled power source in the range of 0–0.5A, with current step changes of 0.01A every 5 s. This allowed us to observe the process of quasistatic heating. The current and voltage drop on a sample with two joints (Figure 4) were measured using HP34401A (Keysight Technologies, Santa Rosa, CA, USA) digital multimeters.

The layer-foil joints were subjected to strength tests. For the purposes of the study, we made a series of 12 identical samples with joint areas of 2 × 2 mm. In order to facilitate assembly of the samples in the vice grip of a CERT system, larger ceramic substrates with dimensions of 12 × 10 mm and Ag strips with dimensions of 2 × 25 mm were used. The sample was pulled to its breaking point to determine the ultimate tensile strength of the joint. The force *F* applied to the sample and the elongation Δ*L* of the sample were measured throughout the tests. The tests were performed for the tangential *F_T_* (6 samples) and normal *F_N_* (6 samples) force components applied to the sample (Figure 5). The results identified the elements of the layer-joint-foil structure that determined its strength.

The tests were carried out using the UMT-2 universal research system by CETR, which has a positioning system with a range of motion of 150 mm, resolution of 0.5 μm and speed of 0.002–10 mm/s. The test stand was equiped with a two-axis sensor with a range of 0.2–25 N and a resolution of 1.0 mN, as well as with a Mecmesin’s Lightweight Double-action Vice Grip (Newton House, Spring Copse Business Park, Slinfold, West Sussex, UK).

### 2.3. Thermo-Electrical Model of Joint

A numerical model was developed to verify the dominant Joule heat source identified by experimental research. The model also enabled us to analyze the heat distribution in the samples. The model was implemented in Comsol Multiphysics software (COMSOL Inc., Burlington, MA, USA). The geometry and model assumptions were made in accordance with the dimensions and parameters of the samples tested at room temperature (Figure 6). This made it possible to compare the results and verify the assumptions used in the model. In accordance with the observations and microscopic measurements, the following dimensions were assumed: thickness of the joint, 20 µm; thickness of the Au layer, 200 nm; thickness of the Ag strips, 35 µm.

The coupled thermal-electric model enabled us to analyze the effects of thermal phenomena including conduction, natural convection, and temperature on the electrical properties of the materials used in the samples. Generally, the simulation modeled the process of heating a sample with two joints, with current supplied by a controlled power source. The current step changes were the same as in the experimental study (current increases of 0.01 A every 5 s).

## 3. Results and Discussion

The various joints (sample A, sample B and sample C), soldered using our method on the Al_2_O_3_ substrate, were subjected to microscopic observations at low magnifications of 5–10×. The images are shown in in Figure 7a, Figure 8a and Figure 9a. The joint area was limited to the size (width) of the Ag (or Au) strip. Almost no flow of solder and flux was observed over the edge of the attached metallic strip. This advantageous feature results from the mechanism of creating a soldered joint according to the proposed method.

The metallographic cross-sections of the joints are shown in Figure 7b, Figure 8b and Figure 9b. The composition of the two-metal alloy created during soldering depends on the components of the alloy, the temperature, and the kinetics of the solidification process. It is well known that the intermetallic phases in a solder joint affect both its electrical and mechanical properties.

Of the joint variants made using our method, special attention should be given to the Ag-In and Au-In alloys. In the proposed soldering process, the temperature in the area of the joint did not exceed 200 °C. It should be noted, however, that the soldered joint is dynamically formed within approx. 10 s, during which the solder ball is initially in a liquid state for 3 s (Figure 1a) and then remains in a liquid state for another 2 s after insertion of the Ag (or Au) strip with flux (Figure 1b). The duration of TLS processes where the same Ag-In and Au-In systems are used [17,18,31] lasted significantly longer. For Ag-In it is 10 min at 210 °C [4,19] and for Au-In, the process lasts 1–10 min at 160–240 °C [16,32]. The metallurgical processes take place simultaneously on the two interfaces, Ag/In and Au/In. It has been established theoretically and experimentally that the kinetics of the formation of intermetallic compounds in such processes depend on diffusion at the interface of the Au/In and Ag/In [32,33]. There are two additional factors that contribute to the formation of the joint. The first is the surface tension of the melted solder ball with a diameter of approx. 0.5 mm, which is much smaller than the joint area (2 × 2 mm). Surface tension is one of the most important factors in alloy formation and for In, reaches the highest value of about 550 mN/m at around 200 °C [34]. The second factor is the wetting of Ag and Au surfaces by the melted solder, depending on the surface energy of each layer. As a result, the molten solder flows immediately between the Ag strip and Au metallization.

Figure 7b shows a cross section of the sample A joint. As can be clearly seen, the whole Au metallization was converted to intermetallic compounds. The alloy between the Au metallization and Ag strip contains up to 31.7 at.% Au and up to 68.3 at.% In, which confirms the formation of an AuIn_2_ single intermetallic phase. The same intermetallic phase has been reported in the temperature range of 180–300 °C when the In thickness is larger than that of Au [35]. The intermetallic compound Ag_2_In was identified between Ag and the liquid In (point 3 on Figure 7b). The reaction Ag-In has been shown to be driven by the diffusion of silver in molten indium, with a parabolic growth constant of only 6.07 × 10^−5^ cm^2^/s at 473 K [36]. This indicates that the Ag_2_In layer produced in our soldering process is only a few micrometers thick.

A cross-sectional microscopic picture of the sample B joint is shown in Figure 8b. Because of the larger amount of In solder involved in the reaction with the Au strip and Au thin layer (a larger In ball was placed on the ceramic substrate during the soldering process), the phase composition of the soldered joint is more complex. A layer with a composition similar to that of AuIn_2_ and a thickness of approx. 5 μm formed at the interface with the Au strip (point 2 on Figure 8b). The presence of the same intermetallic compound was identified at the boundary between the In solder and the Au thin film. The main volume of the solder between these intermetallic phases is In with spherical precipitates of AuIn_2_ and oxides (point 4 on Figure 8b).

A cross-section of the sample C joint is shown in Figure 9b. Although the combination of an Ag/In/Ag thin layer is not recommended (due to the tendency of silver to electromigration), our joining method also worked well. The main phase between the Ag strip and Ag metallization consisted of between 45.4 at.% In (point 3 on Figure 9b) and 66.2 at.% In (point 2 on Figure 8b), with the rest being silver. The phase diagram of the Ag-In binary system [34] manifests two intermetallic phases —γ(Ag_2_In) and φ (AgIn_2_)— and no other phases in the alloy between these two border In contents. In the case of the sample C joint, the solder alloy is a eutectic mixture of these intermetallic compounds.

Studies on the effects of temperature on the resistance of Au layer and Au/In/Ag joint (sample A) were conducted in a wide temperature range, mainly at cryogenic temperatures of 15–300 K. The characteristics of the joints were found to be repeatable and no discontinuities or sudden, unpredictable changes in resistance were observed (Figure 10). This proves the stability of the electrical and mechanical properties of the joints over a wide temperature range. The results showed a linear dependence of resistance on temperature in the range of 35–300 K for both the Au layer and the joint, which is a typical relationship for metals and metal alloys, and which proves the stability of the joint. Stability is a very important property of joints, especially for cryogenic sensors operating in a wide temperature range.

The electrical properties of the joint were determined by the dominant material (silver, which results in low joint resistivity). The temperature coefficient of resistance for the Ag/In/Au joint (*α_J_* = 3.3 × 10^−3^ 1/K) is similar to the coefficient of bulk Ag samples (*α_Ag_Bulk_* = 3.8 × 10^−3^ 1/K) (Figure 9b). This similarity is an effect of the geometry of the tested joint (the large area of the joint in relation to the small thickness), the small size of the intermetallic transition zones and the non-occurrence of metal oxides in the joint. The temperature coefficient of resistance for the Au layer sputtered on the ceramic substrate (*α_Au_* = 2.3 × 10^−3^ 1/K) is lower than the that of bulk Au samples (*α_Au_Bulk_* = 3.7 × 10^−3^ 1/K). The differences in both the layer resistance and the *α* coefficient of the thin (200 nm) vacuum deposited Au layer (Figure 10) in comparison to bulk metal are an effect of the fine-grained, more defective, and heterogeneous structure of the layer.

Due to the good electrical and mechanical properties of the presented joints, we were able to successfully use the proposed method to produce connections in cryogenic sensors [26,30].

Figure 11 shows the results of strength tests on the joints. The strength of the joint to the action of the tangential force component *F_T_* was much greater than its strength to the normal force component F_N_. In this case, a greater elongation Δ*L* was also observed. The change in the elongation of the sample is mainly related to the stretching of the Ag foil and the displacement of the metallic layers in the joint itself. The low value for *F_N_* causes destruction of the joint, due to delamination of the layer from the ceramic substrate. In each of the tested samples, the joint was destroyed by the removal of the metallic layer from the substrate. The mechanical strength was primarily determined by the adhesion of the metallic layer to the surface of the ceramic substrate. It should be noted that mechanical strength can be improved by protecting the joint with resin. In addition, resin reduces the oxidation and corrosion process of the joint that can occur with cyclic heating and cooling to cryogenic temperatures.

In the second part of the electrical tests, the electrical parameters of the samples were measured at room temperature based on analysis of the Joule heat flux density generated in the sample. The Au layer and Au/In/Ag (sample A) joints were integrated on a homogeneous ceramic substrate with high thermal conductivity. Such substrates are often used to build cryogenic sensors, in order to eliminate the temperature gradient in the sensor while ensuring good thermal anchoring and electrical insulation.

The results indicate that the joints were not the dominant heat source in the tested samples. During quasi-static heating, the highest temperatures were in the central part of the samples (Figure 12). This was the effect of the disproportion of the layer and joint resistance. Moreover, the metal leads have an additional function as heat sinks, dissipating the heat generated in the sample. It should be noted that the difference between the joint temperature and the layer temperature was small (~5–8 K), despite the fact that the layer is the dominant part of the sample. The ceramic substrate defined the distribution of heat in the sample.

The experimental results were compared with the results from the numerical model. The results of the simulation confirmed that the Au layer was the dominant heat source in the tested samples (Figure 13). As in the experimental tests, the temperature difference between the joint and the Au layer did not exceed 8 K. The heating process was volumetric, as a result of the quasi-static process of Joule heating and the thermal properties of the substrate.

According to both the experimental results and the numerical model, the heat is distributed in the substrate from the center along the sample (conduction) and to the surroundings (convection). The silver leads, in the form of thin strips, act as heat sinks and dissipate heat to the environment as a result of natural convection. The temperature distribution profiles for all current values are similar. The joints are not significant heat sources.

## 4. Conclusions

In this paper, we have presented an effective method of soldering durable joints to thin metallic layers on ceramic substrates. The main configuration for the connection uses Ag foil joints attached to a thin Au layer using In solder. This configuration was successfully applied in the studied cryogenics structures [26]. The crucial idea of our method is to melt the solder without a flux and place a metal strip with a flux coating on the molten solder in the next step. The process is dynamic and takes only a few seconds and can be used with joints of a similar configuration. In our future research, we will also focus on this issue when using conductive materials important for cryogenics, such as alloys that exhibit low thermal conductivity and a low coefficient of temperature resistance.

The proposed method was also used to make Au foil joints with a thin Au layer using In solder, and Ag joints with a thin Ag layer using In solder. Our method of dynamic soldering (approx. 2 s) produces joints without disturbing the continuity of the layer.

The results of metallographic studies showed the necessary formation of intermetallic layers in the soldered joints. The connections were characterized by low resistance and good mechanical strength across a wide temperature range (15–300 K). Electrical studies and numerical simulations proved the excellent properties of the joints. The proposed method could be used for producing joints in cryogenic sensors.

## Figures and Tables

**Figure 1 sensors-21-04919-f001:**
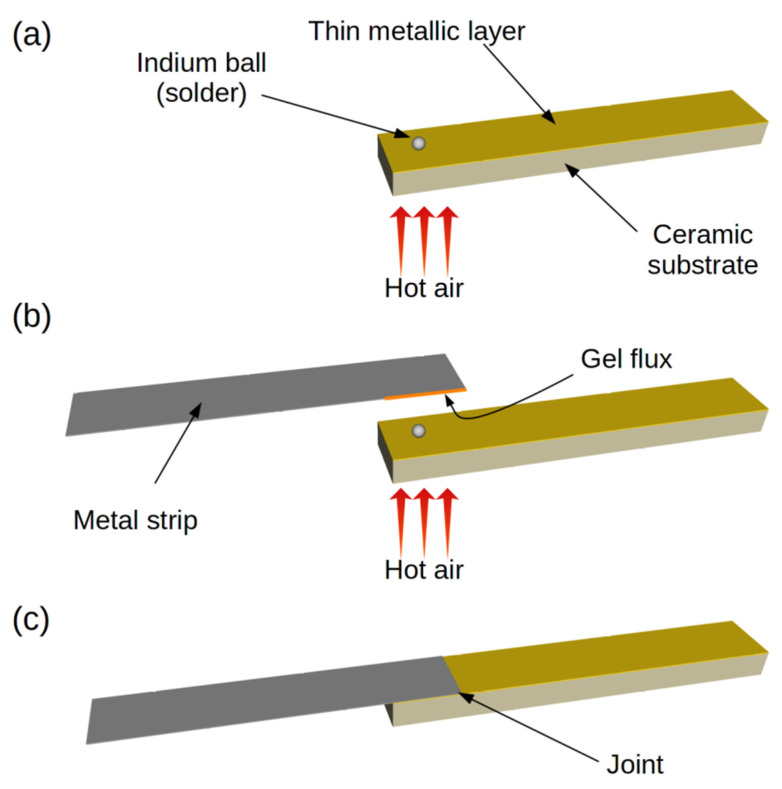
New three-stage process for soldering a metal strip to a thin metallic layer sputtered on a ceramic substrate: (**a**)—Solder melting; (**b**)—Placing a metal strip with a flux over the melted solder; (**c**)—Cooling the joint.

**Figure 2 sensors-21-04919-f002:**
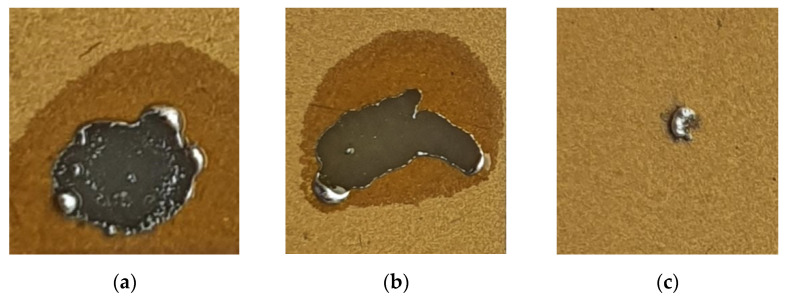
Results of solder melting process: (**a**,**b**) the simultaneous use of In solder and flux; (**c**) the use of In solder without flux.

**Figure 3 sensors-21-04919-f003:**
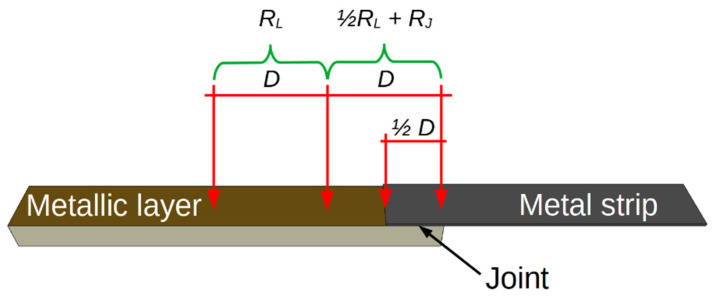
Method of measuring the resistance of joined elements at cryogenic temperatures.

**Figure 4 sensors-21-04919-f004:**
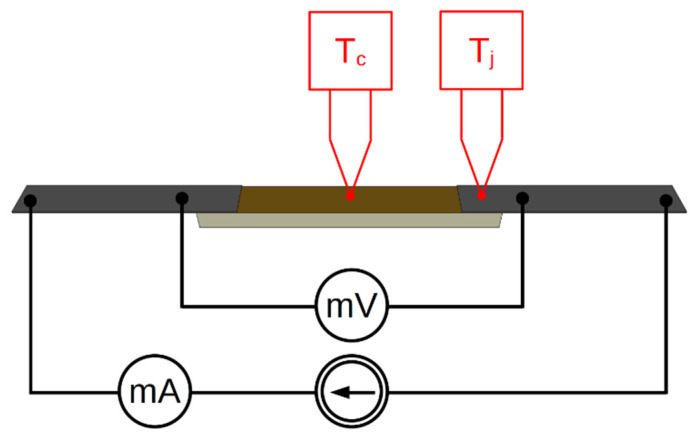
Method of measuring the temperature on the sample surface.

**Figure 5 sensors-21-04919-f005:**
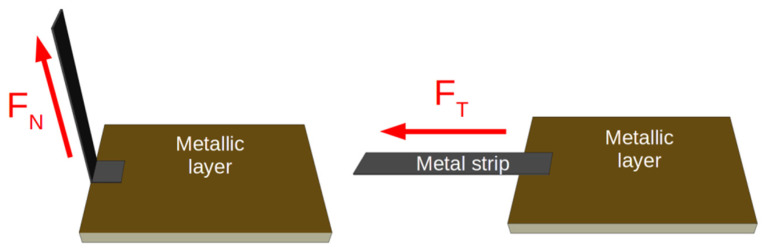
Configurations for strength testing of the joint.

**Figure 6 sensors-21-04919-f006:**
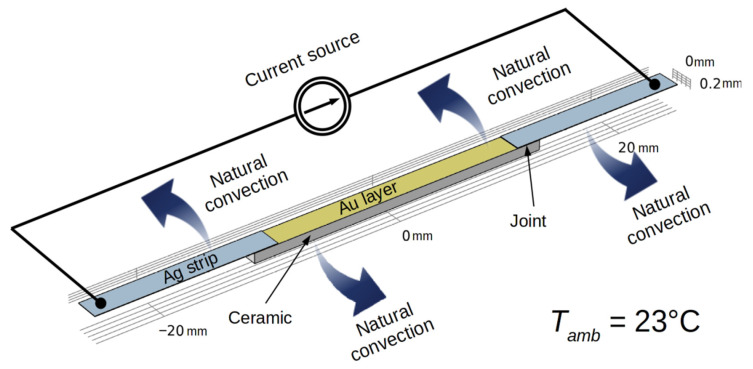
Geometry and boundary conditions of the numerical model of the sample.

**Figure 7 sensors-21-04919-f007:**
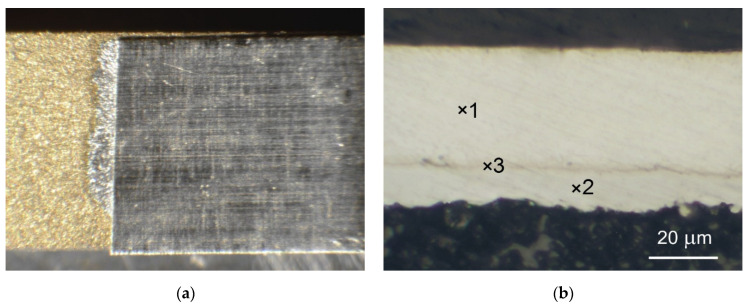
Microscopic pictures of a sample A: (**a**) overall view of joint; (**b**) cross-section of joint with phase composition of different areas: 1—Ag; 2—31.7 at.% Au, 68.3 at.% In; 3—65.8 at.% Ag, 34.2 at.% In.

**Figure 8 sensors-21-04919-f008:**
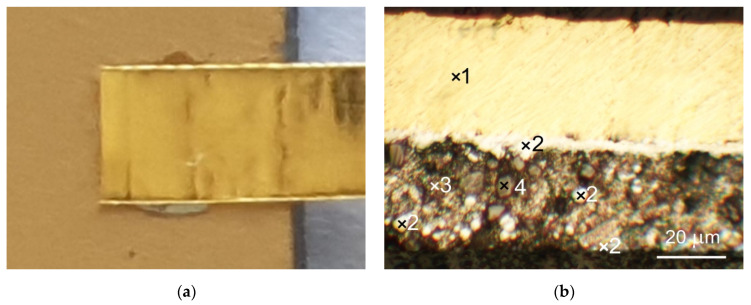
Microscopic pictures of a sample B: (**a**) overall view of joint; (**b**) cross-section of joint with phase composition of different areas: 1—Au; 2—30.7 at.% Au, 69.3 at.% In; 3—In; 4—79.6 at.% In, 20.4 at.% O.

**Figure 9 sensors-21-04919-f009:**
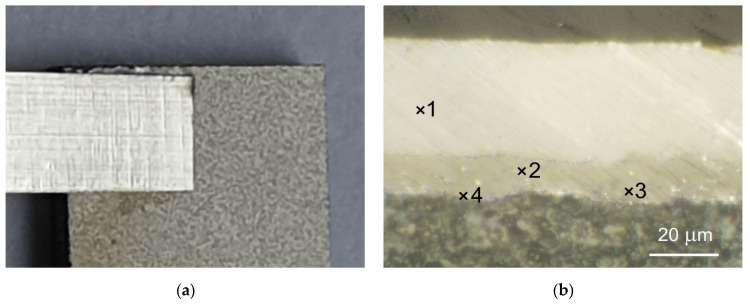
Microscopic pictures of a sample C: (**a**) overall view of joint; (**b**) cross-section of joint with phase composition of different areas: 1—Ag; 2—33.8 at.% Ag, 66.2 at. % In; 3—54.6 at.% Ag, 45.4 at.% In; 4—49.9 at.% Ag, 50.1 at.% In.

**Figure 10 sensors-21-04919-f010:**
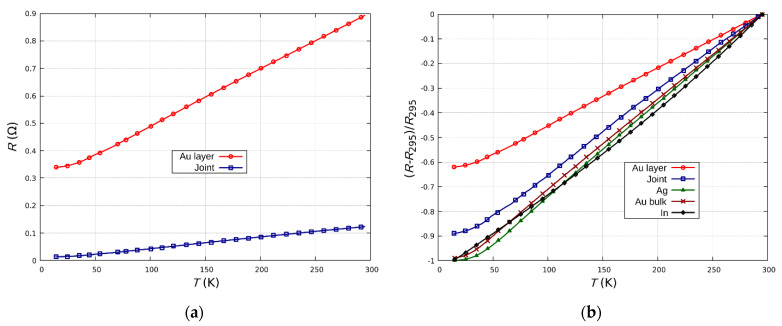
Temperature dependence of resistances (**a**) and relative changes in resistances (**b**) for an Au/In/Ag (sample A) joint and Au layer on a ceramic substrate.

**Figure 11 sensors-21-04919-f011:**
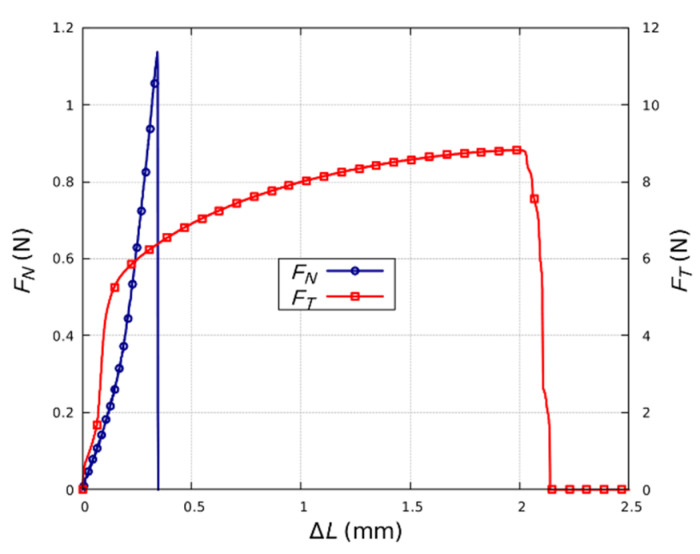
Results of strength testing joints for the tangential component *F_T_* and the normal force component *F_N_*.

**Figure 12 sensors-21-04919-f012:**
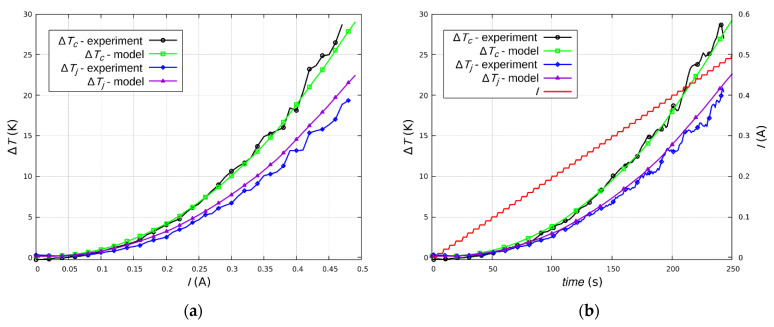
Experimental and simulation temperature changes at the point with the highest temperature (center of the sample T_c_) and at the joint (T_j_): (**a**) temperature-current relationship; (**b**) current and temperature time dependence.

**Figure 13 sensors-21-04919-f013:**
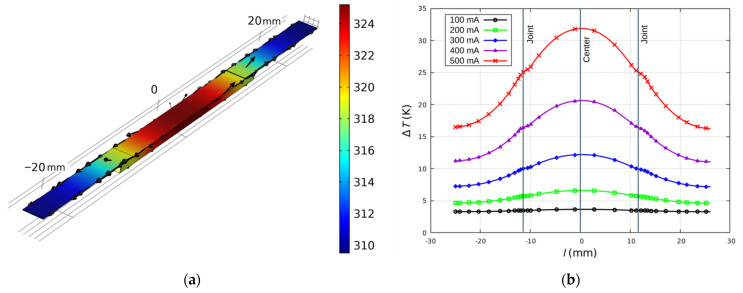
Numerical model results: (**a**) temperature (colors) and heat flux (arrows) distribution for 500 mA; (**b**) temperature profiles along the sample.

**Table 1 sensors-21-04919-t001:** Configuration and dimensions of samples A, B and C.

	Sample A	Sample B	Sample C
Solder	In	In	In
Substrate	Al_2_O_3_ 2 × 25 × 0.8 mm	Al_2_O_3_ 2 × 25 × 0.8 mm	Al_2_O_3_ 2 × 25 × 0.8 mm
Metallic layer	Au 200 nm	Au 200 nm	Ag 200 nm
Lead	Ag strip 2 × 15 × 0.035 mm	Au strip 2 × 15 × 0.05 mm	Ag strip 2 × 15 × 0.035 mm
Flux	Paste flux NC 254 [31]	Paste flux NC 254	Paste flux NC 254

## Data Availability

Not applicable.

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
