# Peer review of "Joining of Electrodes to Ultra-Thin Metallic Layers on Ceramic Substrates in Cryogenic Sensors†"

_sensors, 2021, doi:10.3390/s21144919_

Round 1

Reviewer 1 Report

The authors seem to have presented a promising route towards making electrical connections in materials commonly used for cryogenic sensors. Based on the results presented, I would expect that this technique does exhibit beneficial properties for use in real-world applications. However, the manuscript itself is often too vague and confusing to really understand the scientific progress that was made by this work. It is not clear how this work is different from previous work, and ultimately, there are far too many questions left unanswered. This manuscript is not appropriate for publication, and it needs major revisions/a rewrite before it should be reconsidered. I will provide a few more specific comments and suggestions below.

  1. There are several words missing or awkward English constructions used in the writing of this paper. The authors need to have this manuscript professionally edited/proofed by a service that can fix these issues.
  2. The introduction is a bit too self-serving in my opinion. The authors reference themselves several times, and I don’t believe they have done a complete review of the literature on joining methods. There has been lots of work to understand TLP processes, and what the authors have included seems incomplete and insufficient to explain the motivations and significance of their work.
  3. The authors mention their work in reference 20 and say that “The excellent properties of joints produced using our method have been previously confirmed in the temperature range of 15-300K[20].” It is not clear how this work is different from the work in reference 20, and the work contained in reference 20 seems to be very repetitive and duplicative of the work shown here. It is just not clear how this work is different, and the authors need to be explicit about that because other than the inclusion of a model it seems to repeat the earlier work. This is typically frowned upon by journals, and it would be better for the authors to explain clearly and explicitly how this work is presenting something new and novel that has not been presented before.
  4. It is not clear what scientific question motivated this work. It was not until the methods section page 4 line 169 that the authors explain “The aim of the research was to identify the area of the sample which is the dominant heat source.” If this in fact is the main point of this work, then this needs to be stated very early in the introduction. Also, it is not clear how the work that was done all supports this “aim.” Why do mechanical tests then? How did the microscopy and the intermetallic phases get incorporated into this approach to answering this question? This doesn’t make sense, and the purpose of this manuscript seems to be more than just identifying the dominant heat source. Again, the authors are too vague and have not done a sufficient job of explaining to the reader what the important scientific question is here, and why the answer the authors provide is significant and impactful to the technological community. The text and must be more explicit and concise in answering these questions.
  5. I don’t understand the authors use of the word “solder” in this paper. Is solder the same as the In? Is the solder an intermetallic that forms with In? How are the authors using the word solder in this case because often in other contexts solder and In are not interchangeable. This needs to be consistent throughout.
  6. I don’t understand Figure 3. The authors only show the temperature measurement in one location. However, in Fig 11, the authors show data for T measurement in two locations. This needs to be explained better.
  7. The authors don’t explain how they made an In ball. It seems like this is an important step in the process and this should be explained. The authors talk about a flux, but there is no data on what happens when no flux is used. Is the flux really necessary? What happens if no flux is used? The authors briefly talk about the importance of heating from the ceramic side rather than the metal side, but it is not clear what bad thing happens when the sample is heated from the metal side. Does the metal dewet? Does it oxidize? The individual wetting properties of each metal on the other metals or on the ceramic are not addressed. In general, it is not clear how the authors decided to perform the steps they did to fabricate these joints. There are too many questions, and as a reader, I am left unsatisfied with why the authors did what they did. Much more detail and explanation is required and should be included.
  8. It was very confusing to follow which specific samples the authors made and why data from all of them was not presented in each figure. It seems that the authors made 3 joints: 1) Ag foil/In/Au film, 2) Au foil/In/Au film, and 3) Ag foil/In/Ag film. Why did the authors not make a Ag foil/In/Au film sample? Are the intermetallic phases the authors observed consistent with the phase diagrams? The presentation of EDS results was insufficient. The authors should show maps of things like a eutectic structure. Instead of interchangeably talking about Au or Ag samples, the authors need to be explicit and only describe the 3 samples that they actually made. I was extremely confused and frustrated by the fact that the authors did not organize their presented data and text to explain the specifics of each of the 3 types of samples they made. For example Fig 9 only shows plots for “Au” and “joint”. Where is data for Ag? Where is data for In? It is too incomplete to understand how everything is coming together in these experiments to explain the data. Either more data is needed, or more explanation.
  9. It seems to me that Fig 9 would benefit from pure Ag data, pure In data, and to the extent possible intermetallic data. Otherwise these plots seem out of context and it is not possible to really compare them with all the relevant materials that are present.
  10. For Figure 10, how many samples were tested? What is the standard deviation? Is this just two tests from the same sample construction? Why not test all 3 sample types in both orientations?
  11. I don’t understand Figure 5. While the modeling results show some consistency with the data, the schematic of figure 5 is surely different from the actual experimental setup because Ag foil does not appear on both sides of the joint in reality. Why is the model arranged so differently? I don’t understand why the authors would do it like this?
  12. I don’t understand the difference between the position of the “joint” and the “center”. It seems that the authors have labeled the joint as the edge of the foil (figure 1c), but because of the layered structure of the specimen, there is an interface between the foil and thin film over the entire area near the center. I don’t understand why the authors would think that the greatest temperature rise would be at the edge of the foil? There is no data shown that the “joint” region is any different microstructurally than the center. I am just confused by why it is significant that the center is the dominant heat source? Why is this unexpected?

Ultimately, I am just left with too many questions about what is new here and what the authors accomplished in this manuscript. I would encourage the authors to revise this manuscript and improve it greatly before resubmitting for publication. It is important to remember that average readers will not have any pre-existing knowledge about what you have worked on previously and the state-of-the-art for joining technology. You need to make it easy on the reader to understand what you have discovered and why it is important.

Author Response

Dear Reviewer,
please find attached file with our response.

Reviewer 2 Report

Dear Authors,

I think that the manuscript (MS) must be necessarily improved. I have given some suggestions in the following, but the authors must  check and improve the whole test.

Line 8: If this MS is really an extended version of a conference proceeding, you must refer to it giving all the information about the proceeding/conference.

You continuously used the term method, please make an effort to change it where possible.

Through the whole MS you write about Ag and Au strips or foil and Au layer deposited onto an alumina substrate. You must use the same term to refer to the same element and also make more general the figures.

You must add a table in which you describe the layer onto alumina substrate, the strips and the combinations of soldering naming them in a univocal way.

Abstract

Line 13: sensory is more suitable to “bio-world”. In your case, the use of sensing is more suitable.

Introduction

Lines 27-28: this sentence must be clarified.

Lines 27-28: Pay attention on the use of term challenges (too many times) and problems that are not suitable in the context that you described.

Lines 74-77: At the end of the introduction, you describe in a not sufficient way your work. Explain your present results in more detail.

Materials and Methods

The Materials and Methods appear as a cyclic repetition of concepts. You must rewrite in more synthetic and logic way.

In particular, the first part of the Materials and Methods section is too chatty and verbose (lines-80-100). Please inform the readers more directly and concise of the methods and materials used in your experiments.

Lines 75-83: The sentence is repeated twice.

Section 2.1: about substrates and soldering flux you must put more information. (Line 161: did you buy the ceramic substrates with Au layer? )

Line 103: the method consists in.

Line 123: the placement of the metal foil.

Line 159-161: “Ceramic substrates Al2O3 with dimensions of 25x2x0.8 mm and a 35 μm Ag foil strips with dimensions of 15x2mm were used in the tests. A joint with dimensions of 2x2 mm 160 was made between a 300 nm Au layer deposited on ceramics and a 35 μm metal foil.”

Al2O3 in brackets - a 35 μm Ag foil strips? - foil or strips - metal foil is Ag foil…

Has the Au strips the same dimension as the Ag strips?

The content of Section 3 is in the methods section or not?

Results and Discussion

Line 206 vs line 161: Au layer is 200 or 300 nm?

Lines 228-232: why this comment is put in your results and discussion? It is useless for your MS, or if you want to compare your results with the others you must firstly show yours then the comparison with other Authors.

Lines 240-241: please correct this sentence.

Lines 287-290: please rewrite.

Why you do not include all the results about the Au/In/Au?

Conclusions

At the end of the Conclusion section you need to highlight the advantages of your discoveries for practical application in the research and industrial world. Furthermore you need to indicate future lines of your research.

References

You should add more references, in particular in the introduction and discussion.

Best regards

Author Response

(The authors gave the same response as above.)

Round 2

Reviewer 1 Report

In reading the revised manuscript and the response to my questions it is clear that the authors have attempted to do something to improve the manuscript. There are a few sentences added here and there that do explain some things a little better. However, I would say that the authors have really done the absolute bare minimum in terms of trying to really consider my recommendations and improve the manuscript.

  1. In my opinion, this issue of the thermal properties of their joint still comes out of nowhere. It is true that the authors added information into the introduction that help improve the manuscript, but this sentence that appears on line 108 “Another question, then ,was how to produce a joint with low resistance where the area of the joint is not the dominant heat source.” Nowhere in the introduction do the authors discuss resistive heating of joints and how it is a problem in technology. While I do think that the authors have at least provided the two main issues they are trying to solve early the introduction, which is appropriate, I just don’t think what the authors have provided is particularly well written and it still has questions that are out of context and without appropriate background information.
  2. I think the authors have greatly improved the discussion of their experiments and answered some of my questions about the In balls, and why they did what they did.
    1. I don’t think the authors have cleaned up the use of solder vs indium. I still see that used interchangeably in the manuscript.
    2. I don’t think they authors have addressed q8. I don’t understand their response, and a link to some previous work is not appropriate and does not contribute to improving the manuscript. They have not addressed most of my concerns there.
  3. I had a difficult time understanding the responses after Q8. Sometimes words were missing or typos were present that made it more difficult than it needed to be. For example in the response to Q11 “2 joints were made in tests structure as shown in fig. 3 and fig.” A number is missing that corresponds to the second referenced “fig”. IN my opinion, the authors have not really cleared things up, they just described again what was already in the figures. No attempt was made to improve the language and clarity.

In the end, this manuscript is very average. The authors are not making the kinds of changes that this manuscript needs to make it a good and impactful paper. The authors have simply added the minimum number of words they need to so that I am worn down and won’t complain anymore. I recommend publishing as is with the understanding that it will be an average/mediocre example of joining research. I am not interested in reviewing this again.

Author Response

Dear Reviewer,

please find the attached answers for your comments.

Best regards

Reviewer 2 Report

Dear Author,

please resubmit the manuscript by showing in the text the lines at which you have made your changes.

Please rewrite the lines 13-16.

The reference at the conference proceeding is still lacking (line 8).

Best regards

Author Response

(The authors gave the same response as above.)
